# Artificial Intelligence Based Analysis of Corneal Confocal Microscopy Images for Diagnosing Peripheral Neuropathy: A Binary Classification Model

**DOI:** 10.3390/jcm12041284

**Published:** 2023-02-06

**Authors:** Yanda Meng, Frank George Preston, Maryam Ferdousi, Shazli Azmi, Ioannis Nikolaos Petropoulos, Stephen Kaye, Rayaz Ahmed Malik, Uazman Alam, Yalin Zheng

**Affiliations:** 1Department of Eye and Vision Science, Institute of Life Course and Medical Sciences, University of Liverpool, Liverpool L7 8TX, UK; 2Department of Cardiovascular & Metabolic Medicine, Institute of Life Course and Medical Sciences, University of Liverpool, Liverpool L7 8TX, UK; 3Institute of Cardiovascular Science, University of Manchester, Manchester M13 9PL, UK; 4Manchester Diabetes Centre, Manchester Foundation Trust, Manchester M13 0JE, UK; 5Department of Medicine, Weill Cornell Medicine-Qatar, Doha 24144, Qatar; 6St Paul’s Eye Unit, Royal Liverpool University Hospital, Liverpool L7 8XP, UK; 7Liverpool Centre for Cardiovascular Science, Liverpool Heart and Chest Hospital, Liverpool L14 3PE, UK

**Keywords:** artificial intelligence, corneal confocal microscopy, diabetic peripheral neuropathy

## Abstract

Diabetic peripheral neuropathy (DPN) is the leading cause of neuropathy worldwide resulting in excess morbidity and mortality. We aimed to develop an artificial intelligence deep learning algorithm to classify the presence or absence of peripheral neuropathy (PN) in participants with diabetes or pre-diabetes using corneal confocal microscopy (CCM) images of the sub-basal nerve plexus. A modified ResNet-50 model was trained to perform the binary classification of PN (PN+) versus no PN (PN−) based on the Toronto consensus criteria. A dataset of 279 participants (149 PN−, 130 PN+) was used to train (*n* = 200), validate (*n* = 18), and test (*n* = 61) the algorithm, utilizing one image per participant. The dataset consisted of participants with type 1 diabetes (*n* = 88), type 2 diabetes (*n* = 141), and pre-diabetes (*n* = 50). The algorithm was evaluated using diagnostic performance metrics and attribution-based methods (gradient-weighted class activation mapping (Grad-CAM) and Guided Grad-CAM). In detecting PN+, the AI-based DLA achieved a sensitivity of 0.91 (95%CI: 0.79–1.0), a specificity of 0.93 (95%CI: 0.83–1.0), and an area under the curve (AUC) of 0.95 (95%CI: 0.83–0.99). Our deep learning algorithm demonstrates excellent results for the diagnosis of PN using CCM. A large-scale prospective real-world study is required to validate its diagnostic efficacy prior to implementation in screening and diagnostic programmes.

## 1. Introduction

Diabetic peripheral neuropathy (DPN) can lead to neuropathic pain, foot ulcers, amputation, and premature mortality. Early screening and diagnosis are key to implement risk factor reduction to prevent or delay the progression of DPN [1]. Corneal confocal microscopy (CCM) is a rapid non-invasive ophthalmic imaging technique that enables quantification of the corneal sub-basal nerve plexus to detect early DPN [2,3]. Artificial intelligence (AI) and AI-based deep learning algorithms (DLAs) have been utilized to accurately diagnose DPN [4,5,6,7]. We previously developed an AI-based DLA for the segmentation of corneal nerves [7]. More recently, we developed an AI-based DLA [4] utilizing end-to-end classification to analyze CCM images and classify healthy controls and people with diabetes or pre-diabetes with and without peripheral neuropathy (PN). In a clinical diabetes outpatient/screening environment, rapid, reproducible tests are required to define the presence or absence of DPN (binary classification). We hypothesized that an AI-based DLA with a binary classification would outperform our previously published multi-class classification approach. We propose a novel DLA for the binary classification of peripheral neuropathy (PN+) versus no PN (PN−) in people with diabetes and pre-diabetes.

## 2. Materials and Methods

### 2.1. Dataset and Participants

The dataset (ENA group, University of Manchester, UK) consisted of CCM images from 279 participants (type 1 diabetes with PN− [*n* = 49], type 1 diabetes with PN+ [*n* = 39], type 2 diabetes with PN− [*n* = 74], type 2 diabetes with PN+ [*n* = 67], pre-diabetes with PN− [*n* = 26], and pre-diabetes with PN+ [*n* = 24]) using a standard validated protocol. Of the 279 participants, 149 had PN− and 130 had PN+ and their clinical characteristics and neuropathy data can be found in our previously published study [4]. PN was defined according to the Toronto consensus on diabetic neuropathy [8] and pre-diabetes was defined using WHO/ADA criteria. Additional causes of peripheral neuropathy were excluded based on a comprehensive medical and family history and blood tests (immunoglobulins, anti-nuclear antibody, vitamin B12 levels, thyroid function tests). Ethical and institutional approvals were obtained before the participants completed the scientific protocol including CCM imaging which was conducted as a part of 4 longitudinal cohort studies (North Manchester Research Ethics committee, Ref: 09/H1006/38; North West-Greater Manchester East, Ref: 14/NW/0093; Central Manchester Local Research Ethics Committee, Ref: 07/H1006/68; and National Research Ethics Service committee North West, Ref: 08/H1004/1). All individuals provided informed valid consent prior to participation. The research adhered to the tenets of the Declaration of Helsinki.

### 2.2. Algorithm Architecture and Implementation

The AI-based DLA was developed by modifying our previously devised DLA [4] which utilized the pre-trained residual neural network ResNet-50 [9] as the backbone network to perform the classification task. An additional dropout layer with a dropout rate of 0.4 was added to increase the algorithm’s generalisability. One additional fully connected layer with two output neurons was also added to predict the final binary classification results. Data augmentation techniques such as random rotation (0–30 degrees), and horizontal flips with a probability of 0.3 were used during the training process to avoid overfitting problems and increase the algorithm’s generalisability. Stochastic gradient descent with a momentum of 0.9 was adopted to optimize the algorithm. The algorithm was trained for 300 epochs with a learning rate of 0.0001 and a step decay rate of 0.999 every 30 epochs for steady parameter optimization during training. Bilinear interpolation was used to reduce the picture from 384 × 384 to 224 × 224 pixels. The image channel was increased from 1 to 3 by replicating along the channel, turning the single-channel grayscale CCM images into three-channel ‘colourscale’ images. This allowed the images to fit into the ResNet-50 model, pre-trained on ImageNet [10], to limit overfitting and remedy the poor generalisation ability resulting from the limited dataset size used. The pixel value was scaled into [0–1] by dividing by 255, and then normalised into the range of [−1, 1] by using a mean value of 0.5 and a standard deviation value of 0.5 for three channels. Such pixel-wise normalisation can prevent gradient vanishing or exploding during training, resulting in a steady training process.

The algorithm was developed and evaluated on the CCM images from 279 participants (149 PN−, 130 PN+), utilizing one image per participant. Random seeds were automatically generated to split the dataset into training, validation, and testing sets, containing 200 (112 PN−, 88 PN+), 18 (8 PN−, 10 PN+), and 61 images (29 PN−, 32 PN+), respectively.

### 2.3. Performance Evaluation

Performance was evaluated by generating a confusion matrix, a table displaying the true classifications against the classifications predicted by the AI-based DLA. Based on this information, a classification report was generated with the performance metrics: sensitivity, specificity, and area under the curve (AUC). Attribution-based methods of Gradient-weighted Class Activation Mapping (Grad-CAM) and Guided Grad-CAM [11] were used to generate saliency maps to provide explainability to the AI-based DLA decision making.

## 3. Results

The confusion matrix generated after the trained AI-based DLA had classified the test dataset (*n* = 61) is displayed in Table 1. Of the PN− images in the test set (*n* = 29), 27 were correctly detected, and 2 were misclassified as PN+. Of the PN+ images in the test set (*n* = 32), 29 were correctly detected and 3 were misclassified as PN−. The AI-based DLA had a sensitivity of 0.91 (95% CI: 0.79–1.0), a specificity of 0.93 (95% CI: 0.83–1.0), and an AUC of 0.95 (95% CI: 0.83–0.99) in detecting PN+ (Figure 1).

Figure 2 demonstrates representative Grad-CAM and Guided Grad-CAM saliency map images from subjects with correctly detected PN− and correctly detected PN+. Corneal nerves were highlighted in both correctly detected PN− and PN+ images, particularly evident in the Guided Grad-CAM images. Based on previous studies [2], these were appropriate features for the AI-based DLA to be utilizing for classification. However, due to the post-hoc nature of these explainability methods they are less informative in explaining individual decisions, with interpretations risking confirmation bias and misuse [12].

## 4. Discussion

In this study, we propose an accurate AI-based DLA for the binary classification of patients with pre-diabetes and diabetes into those with and without PN through analysis of their CCM images. We have validated the algorithm’s ability to accurately perform this binary classification and demonstrate superior performance to our previous AI-based DLA [4]. Our AI-based DLA provides several potential benefits: (1) achieves excellent results in the classification of PN+ vs. PN− (AUC 0.95); (2) provides the clinically relevant binary classification outcome of PN+ vs. PN− with utility in a diabetes outpatient setting; (3) performs classification without the need for expert annotation, remedying operator bias or automated segmentation with a reliance on pre-determined morphological parameters; and (4) provides rapid automated classification of CCM images which can enable its use in the screening of DPN in a future bespoke diabetic neuropathy screening service.

Several studies have demonstrated successful results in the classification of DPN using AI-based DLAs to analyse CCM images [4,5,6,7]. Scarpa et al. [6] utilized three non-overlapping images of each eye per participant to classify them as a control or PN+, achieving a sensitivity of 0.98, a specificity of 0.96, and accuracy of 0.97. Williams et al. [7] developed a DLA to quantify the corneal nerve morphometrics of CCM images, demonstrating a sensitivity of 0.68, a specificity of 0.87, and an AUC of 0.83 in the classification of PN+. Salahouddin et al. [5] used a U-net DLA, achieving a sensitivity of 0.92, a specificity of 0.8, and an AUC of 0.95 in the classification of PN− vs. PN+. Our previous AI-based DLA performed a multi-class classification between healthy controls, PN−, and PN+, demonstrating a sensitivity of 0.83, a precision of 1.0, and F1-score of 0.91 in the classification of PN+ [4]. Our binary classification model outperformed our multi-class classification model as expected, while also demonstrating a more clinically relevant classification to a clinical diabetes outpatient/screening environment in classifying the presence or absence of DPN.

The life expectancy of people with diabetes is shortened by up to 15 years, with 75% dying prematurely of diabetes-related complications [13]. DPN affects at least 50% of people with diabetes and is the major driving factor for foot ulceration and subsequent lower limb amputation [14]. Around 50% of patients who develop a diabetic foot ulcer die within 5 years [13]. Neuropathic pain is a major feature of DPN, affecting around one-third of all patients and associated with increased morbidity [15]. In the natural history of DPN, small nerve fibre damage precedes large fibre damage, of which the former cannot be detected using current tests such as monofilament insensitivity and loss of vibration perception. Therefore, there is an unmet clinical need to accurately detect early sub-clinical DPN [16]. Indeed, studies have demonstrated an excess of PN in pre-diabetes [17] and the 2017 American Diabetes Association (ADA) position statement on diabetic neuropathy advised screening for pre-diabetes in patients with symptoms of PN [18]. Rapid binary classification into those with and without PN is key to implementing screening for DPN [19]. AI algorithms have already clearly demonstrated rapid and clinically equivalent results to human graders in detecting high-risk diabetic retinopathy [20]. Current (National Institute for Health and Care Excellence [NICE] advocated) methods of detecting DPN are crude and demonstrate poor sensitivity (~50%) and an inability to detect early DPN [13,19]. The 10 g monofilament and 128 Hz tuning fork only detect those at high risk of foot ulceration, when DPN is well-established and irreversible [21]. Therefore, more sensitive diagnostic tests of DPN are required. 

Our group has pioneered the use of CCM, which images small nerve fibres in the cornea. A wealth of published data has demonstrated CCM to be a valid and accurate endpoint for the diagnosis of early and more advanced DPN [2,7,22,23,24,25,26,27,28]. We, and others, have demonstrated that CCM has comparable diagnostic utility to intra-epidermal nerve fibre density in a skin biopsy [2,25]. Indeed, CCM has high sensitivity to detect early DPN and has been proposed as an objective and reliable biomarker for screening and diagnosis programmes [19]. 

AI-based diagnostics utilizing CCM, a rapid ophthalmic imaging technique, would therefore enable screening for DPN alongside diabetic retinopathy. Our AI-based DLA significantly outperforms current clinical methods for the screening and diagnosis of DPN [29]. The DLA was trained and tested on a limited sized dataset (*n* = 279 participants) and requires validation in a large-scale prospective study to test its performance in real-world clinical deployment. This is crucial, as previous work in diabetic retinopathy has demonstrated that AI systems may have a lower performance in clinical practice compared to in-lab validation [30]. Furthermore, the potential health economic impact of introducing the DLA within a DPN screening programme will require determination through health-economic models. Further studies should also develop AI-based DLA that can identify participants at-risk of developing DPN.

## 5. Conclusions

Our AI-based DLA demonstrated excellent diagnostic ability for DPN and therefore has the potential to be used in screening programmes for DPN. Further large-scale real world clinical validation is required. 

## Figures and Tables

**Figure 1 jcm-12-01284-f001:**
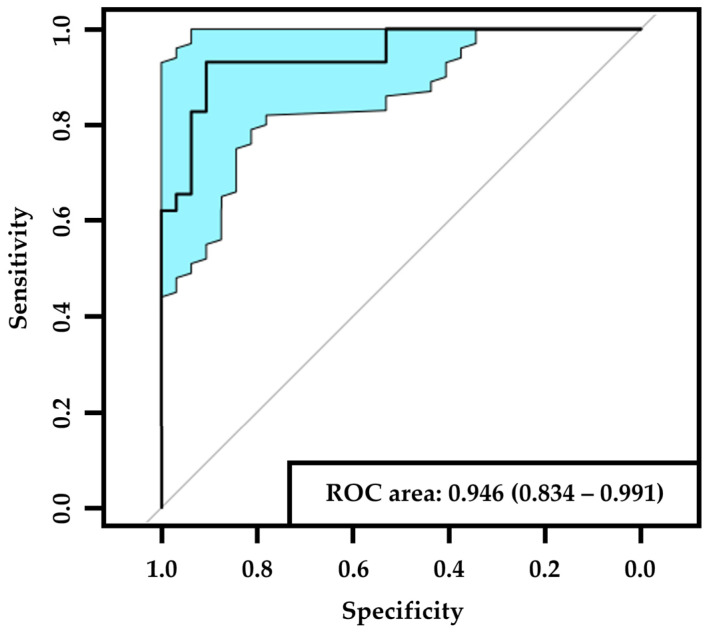
ROC (receiver operating characteristic) curve of PN+. The black line corresponds to the ROC curve and the blue area corresponds to the 95% confidence interval. List of abbreviations: PN+—peripheral neuropathy.

**Figure 2 jcm-12-01284-f002:**
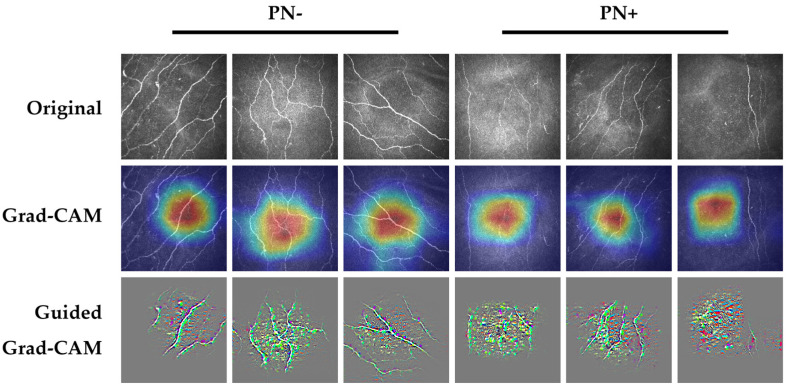
Example saliency map images from correctly detected PN− (columns 1,2,3) and PN+ (columns 4,5,6). Highlighted areas within the Grad-CAM and Guided Grad-CAM images demonstrate the areas in the image which impacted the classification decision most. For Grad-CAM images, the areas of the image highlighted in red had the most impact on the classification decision, followed by orange, yellow, green, light blue, and dark blue. Top row, original images; middle row, Grad-CAM images; bottom row, Guided Grad-CAM images. List of abbreviations: PN+—peripheral neuropathy; PN−—no peripheral neuropathy.

**Table 1 jcm-12-01284-t001:** Confusion matrix of the AI-based DLA in participants with diabetes and pre-diabetes with and without peripheral neuropathy.

		Predicted Class	
		PN−	PN+
**True Class**	**PN−**	27	2
	**PN+**	3	29

List of abbreviations: PN+—peripheral neuropathy; PN−—no peripheral neuropathy.

## Data Availability

The datasets generated during and/or analyzed during the current study are available from the corresponding author on reasonable request after the completion of secondary analyses and further algorithm development.

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
