# Peer review of "Artificial Intelligence Based Analysis of Corneal Confocal Microscopy Images for Diagnosing Peripheral Neuropathy: A Binary Classification Model"

_jcm, 2023, doi:10.3390/jcm12041284_

Round 1
Reviewer 1 Report
In this report the authors present the results of artificial intelligence based DLA for identification of diabetic changes in SBNP using CCM images.
The use of CCM to detect, follow and monitor diabetic peripheral neuropathy was widely reported and its role is indisputable (please cite Roszkowska AM et al. Corneal nerves in diabetes—The role of the in vivo corneal confocal microscopy of the subbasal nerve plexus in the assessment of peripheral small fiber neuropathy. Survey of Ophthalmol. 2021 May-Jun;66(3):493-513.
As the AI and machine learning is developing rapidly to improve the diagnosis in medicine, the present paper is of great actuality.
Hovewer, I believe that the diabetes duration, severity and the patients age and sex should be considered and the study groups at this stage of DLA development should be homogeneous.
These informations are missing in the study.
Reviewer 2 Report
It is clear that the authors know a good deal with artificial intelligence helping of current imaging technology for diagnosing peripheral neuropathy. This manuscript addressed scientific and practical method of corneal confocal microscopy (CCM) images using AI-based deep learning algorithm (DLA) with binary classification showed outperformance than current multiple classification approaches on diagnosing peripheral neuropathy.
Comment 1: In manuscript, the results showed an accurate AI-based deep learning algorithm when analyzing patients with pre-diabetes and diabetes CCM images. But because it is not a new finding that CCM help with diagnosing peripheral neuropathy, what the details of benefit when using this more advanced DLA with binary classification CCM approach than others which with multiple classification approaches? Can they be comparable from any results? It will be perfect to show more data (images or result), even more discussion on that. For example, in Table1, The AI-based DLA had a sensitivity of 0.91 (95% CI: 0.79–1.0), specificity of 0.93 (95% CI: 0.83–1.0) and an AUC of 0.95 (95% CI: 0.83–0.99) in detecting PN+. This is relatively high-sensitivity & high-specificity performance, which is no doubt about it. What about the performance of other approaches? Has this new algorithm really done better job than others (multiple classification)? We don’t know if no comparison data showing.
Comment 2: Fig2 shows a color channel in the middle row called Grad-CAM. It may need to put more information to explain what these colors mean in these images. Otherwise, it is confusing that what they are, what the colors stand for? How these results help to detect nerves.
Comment 3: Discussion needs to be polished by adding more explaining benefit of this special DLA on CCM. Current version just grabbing small piece that audience barely can get the advantage of this high-sensitivity & high-specificity AI-based DLA.
Round 2
Reviewer 2 Report
Thank you for the responses. I personally accept the revised manuscript and agree to be published.